# A High-Performance Computing Implementation of Iterative Random Forest for the Creation of Predictive Expression Networks

**DOI:** 10.3390/genes10120996

**Published:** 2019-12-02

**Authors:** Ashley Cliff, Jonathon Romero, David Kainer, Angelica Walker, Anna Furches, Daniel Jacobson

**Affiliations:** 1Bredesen Center for Interdisciplinary Research and Graduate Education, University of Tennessee Knoxville, Knoxville, TN 37996, USA; ashley.cliff4@gmail.com (A.C.); jromero1208@gmail.com (J.R.); walkeram@ornl.gov (A.W.); furchasak@ornl.gov (A.F.); 2Oak Ridge National Laboratory, Oak Ridge, TN 37830, USA; kainerd@ornl.gov

**Keywords:** Random Forest, Iterative Random Forest, Gene Expression Networks, high-performance computing, X-AI-based eQTL

## Abstract

As time progresses and technology improves, biological data sets are continuously increasing in size. New methods and new implementations of existing methods are needed to keep pace with this increase. In this paper, we present a high-performance computing (HPC)-capable implementation of Iterative Random Forest (iRF). This new implementation enables the explainable-AI eQTL analysis of SNP sets with over a million SNPs. Using this implementation, we also present a new method, iRF Leave One Out Prediction (iRF-LOOP), for the creation of Predictive Expression Networks on the order of 40,000 genes or more. We compare the new implementation of iRF with the previous R version and analyze its time to completion on two of the world’s fastest supercomputers, Summit and Titan. We also show iRF-LOOP’s ability to capture biologically significant results when creating Predictive Expression Networks. This new implementation of iRF will enable the analysis of biological data sets at scales that were previously not possible.

## 1. Introduction

Due to innovation in the areas of genome sequencing and ’omics analysis, biological data is entering the age of big data. As opposed to other fields of research, biological data sets tend to have large feature quantity, but much smaller sample counts, such as in a GWAS population where there are typically hundreds to thousands of genotypes, but millions of SNPs. The number of independent features of biological systems is large, and would require many lifetimes of the entire scientific community to sufficiently study [1]. The ability to determine which features are influential to a particular phenotype, be it SNPs, gene expression, or interactions between multiple molecular pathways, is essential in reducing the full feature space to a subset that is feasible to analyze.

Many methods exist to determine feature importance and feature selection, such as Pearson Correlation, Mutual Information (MI), Sequential Feature Selection (SFS) [2], Lasso, and Ridge Regression. Random Forest [3] is a commonly used machine learning method for making predictions, and while not classically defined as a feature selection method, it is useful in scoring feature importance. During the training phase, decision trees are built, where a subset of features is examined at each decision point and the one that best divides the data is chosen. Feature importance is calculated for each feature based on its location and effectiveness within the tree structures. By using random subsets of the training data for each tree and considering random features for each decision point, Random Forest prevents over fitting. Because of the nature of decision trees, the importance of any chosen feature is inherently conditional on the features that were chosen previously. In this way the Random Forest can account for some of the interconnected dependencies that occur in biological systems. As a non-linear model, Random Forest has been applied to a range of biological data problems, including GWAS, genomic prediction, gene expression networks and SNP imputation [4].

Iterative Random Forest [5] (iRF) is an algorithmic advancement of Random Forest (RF), which takes advantage of Random Forests ability to produce feature importance and produces a more accurate model by iteratively creating weighted forests. In each iteration, the importance scores from the previous iteration are used to weight features for the current forest. Until now, iRF was implemented solely as an R package [6]. While useful for small projects, it was not designed for big data analysis. This paper describes the process of implementing a high-performance computing-enabled iRF, using MPI (Message Passing Interface) [7]. This new implementation enabled the creation of Predictive Gene Expression Networks with 40,000 genes and quickly completed the feature importance calculations for 1.7 million *Arabidopsis thaliana* SNPs in relation to a gene expression profile, as part of a genome-wide explainable-AI-based eQTL analysis.

## 2. Materials and Methods

### 2.1. Random Forest and Iterative Random Forest Methods

The base learner for the Random Forest (RF) and Iterative Random Forest (iRF) methods is the decision tree, also known as a binary tree. A decision tree starts with a set of data: samples, features, and a dependent variable. The goal is to divide the samples, through decisions based on the features, into subsets containing homogeneous dependent variable values, generally following the CART (Classification and Regression Trees) method [8], where each decision divides one node into two child nodes. This is a greedy algorithm which will continue to divide the samples into child nodes, based on a scoring criterion, until a stopping criteria is met. Decision trees are weak learners and tend to over-fit to the data provided.

A Random Forest is an ensemble of decision trees. However, the trees in a Random Forest differ from standard decision trees in that each tree starts with a subset of the samples, chosen via random sampling with replacement. Also differing from standard decision trees, the features being considered at each node are a random subset, with the number of features provided by a parameter. Once a forest has been generated, the importance of each feature can be calculated from node impurity, such as the Gini index for classification or variance explained for regression, or permutation importance. Unlike a single decision tree, a Random Forest is a strong learner, because it averages many weak learners and avoids putting too much weight on outlier decisions. The number of trees in a forest is a parameter that is chosen by the user and influences the accuracy of the model.

Iterative Random Forest expands on the Random Forest method by adding an iterative boosting process, producing a similar effect to Lasso in a linear model framework. First, a Random Forest is created where features are unweighted and have an equal chance of being randomly sampled at any given node. The resulting importance scores for the features are then used to weight the features in the next forest, thus increasing the chance that important features are evaluated at any given node. This process of weighting and creating a new Random Forest is repeated *i* times, where *i* is set by the user. Due to the ability to easily follow the decisions that these models make, they have been deemed explainable-AI (X-AI) [1], which differs from many standard machine and deep learning methods.

For both Random Forest and Iterative Random Forest, the total number of features, samples, trees, and iterations (for iRF) all influence run time to differing degrees. Most of the computation time is spent creating the decision trees, so run time scales with the number of trees. The number of samples influences the number of decisions within a tree needed to divide the data into homogeneous subsets. This influences the number of nodes created in a tree, and thus the run time per tree. Furthermore, the number of features influences the amount of time required to find the feature that best divides the samples at any given node. Finally, for iRF, a whole new forest must be generated for each iteration, though the run time for subsequent forests tends to diminish as many feature weights are set to zero. The number of iterations allows the user to find a balance between over- and under-parameterization of the model, by progressively eliminating features.

### 2.2. Implementation of iRF in C++

We used Ranger [9], an open source Random Forest implementation in C++, as the core of our iRF implementation. Ranger already implements the decision tree and forest creation aspects, but implements neither the communication necessary for running on multiple compute nodes of a distributed HPC system, nor the iterative aspect of iRF.

Within our implementation, each decision tree is initialized with a random subset of the data sample vectors and is then built in an independent process. Groups of trees (sub-forests) are built on compute nodes and then sent to a master compute node that aggregates them into a full Random Forest. This is done by giving each sub-forest a randomly generated seed number which determines the random subset of data each tree in a sub-forest uses, allowing for a higher likelihood of unique random data subsets on each tree. Once in the form of a single Random Forest, Ranger’s functions for forest analysis are used, including feature importance aggregation.

On a distributed system, where parts of the forest are created on different compute nodes, the aggregation of results requires most of the inter-node communication. This process relies on MPI (the Message Passing Interface), an internode-communication standard for parallel programming, and Open-MPI [10], an open source C library containing functions that follow the MPI standard.

To implement *i* iterations, the forest creation and aggregation is performed *i* times. After the completion of each iteration, the feature weights are written to a file. At the beginning of the next iteration, this file is read into an array that is used to create the weighted distribution from which features are sampled during the decision process at each node on each tree. This process, while potentially slow due to the file I/O, uses the preexisting functionality of weighted sampling from Ranger.

### 2.3. iRF-LOOP: iRF Leave One out Prediction

Given a data set of *n* features and *m* samples, iRF Leave One Out Prediction (iRF-LOOP) starts by treating one feature as the dependent variable (Y) and the remaining *n* − 1 features as predictor matrix (X) of size *m x n* − 1. Using an iRF model, the importance of each feature in X, for predicting Y, is calculated. The result is a vector, of size *n*, of importance scores (the importance score of Y, for predicting itself, has been set to zero). This process is repeated for each of the *n* features, requiring *n* iRF runs. The *n* vectors of importance scores are concatenated into an *n x n* importance matrix. To keep importance scores on the same scale across the importance matrix, each column is normalized relative to the sum of the column. The normalized importance matrix can be viewed as a directional adjacency matrix, where values are edge weights between features. See Figure 1 for a diagram of this process. Due to the nature of iRF, the adjacency matrix is not symmetric as feature A may predict feature B with a different importance than feature B predicts feature A.

### 2.4. Big Data: Showing the Scale of iRF with *Arabidopsis thaliana* SNP Data

A typical use case of iRF, with a matrix of features and a single target vector of outcomes, becomes comparable to an X-AI-based eQTL analysis, when the matrix of features is a set of single nucleotide polymorphisms (SNPs) and the dependent variable vector is a gene’s expression measured across samples. This analysis determines which set(s) of SNPs are important to variation in the gene’s expression.

We obtained *Arabidopsis thaliana* SNP data from the Weigel laboratory at the Max Planck Institute for Developmental Biology, available at https://1001genomes.org/data/GMI-MPI/releases/v3.1/. We filtered the SNPs using bcftools 1.9 [11] to keep only those that were biallelic, had a minor allele frequency greater than 0.01, and had less than ten percent missing data across the population. This resulted in a set of 1.71 million SNPs for 1135 samples, from the original 11.7 million SNPs.

We obtained *Arabidopsis thaliana* expression data [12] from 727 samples and 24,175 genes. Of these 727 samples, 666 samples were also present in the SNP data set. The vector of gene expression values for gene AT1G01010 for those 666 samples was used as the dependent variable for iRF. The feature set was the full set of 1.71 million SNPs, for the same 666 samples. While this is not many samples, this is clearly many features.

The C++ iRF code was run using these data as input with five iterations, each generating a forest containing 1000 trees. The number of trees was chosen as a value close to the square root of the number of features (a common setting for this parameter), where each feature has a 95% chance of being included in the feature subset within the first two layers in at least one tree. This helps to guarantee that all features are considered at least once across an iteration. HPC node quantities of 1, 2, 5, and 10 were used to show run time changes as the amount of resources increases.

Due to the large number of SNPs, the full data could not fit in memory on a standard laptop, for use in the R iRF program. Instead, small subsets of features of sizes 2000, 1000, 500, 100 and 50 features were run, each using the full 666 samples and the same AT1G01010 gene expression dependent variable. Each feature set was run three times and averaged to account for the inherent stochasticity in the algorithm. Only one run was performed for each parameter set on Summit with the full data set due to limited compute time availability.

### 2.5. Using iRF-LOOP to Create Predictive Expression Networks

Given a matrix of gene expression data, there are a multitude of approaches for inferring which genes potentially regulate the expression of other genes, ranging in complexity from pairwise Pearson Correlations to advanced methods such as Aracne [13], Genie3 [14], and dynamic Bayesian networks (DBNs) [15]. Due to the large number of features and the complexity of the interactions between them, Random Forest-type approaches are well suited to this task.

We applied iRF-LOOP to a matrix of gene expression data measured in 41,335 genes across 720 genotypes of *Populus trichocarpa* (Black cottonwood). The RNAseq data [16] from were obtained from the NCBI SRA database (SRA numbers: SRP097016– SRP097036; www.ncbi.nlm.nih.gov/sra). Reads were aligned to the *Populus trichocarpa* v.3.0 reference [17]. Transcript per million (TPM) counts were then obtained for each genotype, resulting in a genotype–transcript matrix, as referenced in [18]. The adjacency matrix resulting from iRF-LOOP represents a Predictive Expression Network (PEN) where a directed edge (AB) between and two genes (A and B) is weighted according to the importance of gene A’s expression in predicting gene B’s expression, conditional on all other genes in the iRF model. We removed zeros and produced four thresholded networks, keeping the top 10%, 5%, 1%, and 0.1% of edge scores, respectively.

To determine the biological significance of the PEN produced by the iRF-LOOP, we compared each of the four thresholded networks to a network of known biological function, created from Gene Ontology (GO) annotations. GO is a standardized hierarchy of gene descriptions that captures the current knowledge of gene function. It is accepted that there is both missing information and some error, nevertheless it is useful as a broad truth set for large sets of genes. We calculated scores for each network by intersecting with the GO network. We then evaluated the probability of achieving such scores relative to random chance by creating null distributions of scores produced by random permutation of the iRF-LOOP networks and calculated *t*-statistics for each threshold.

We created a Populus trichocarpa GO network for the Biological Process (BP) GO terms, using annotations from [19]. Genes are connected if they share one or more GO terms. Due to the hierarchical nature of the Gene Ontology, a term shared by only a few genes, indicates a very detailed and specific association, where the associations shared by many genes are generally broad categorizations. The edge weight between two genes equals 1/(n−1), where *n* is the number of genes with the shared term. This weighting attempts to balance the scoring metric so that correctly identifying rare edges is not out weighted by simply capturing many generic GO terms. Genes that share a GO term create connected sets of genes, where each gene has an edge with each other gene in the set. A predicted network’s intersect score is a summation of all GO edge weights from edges that appear in both the GO and predicted networks. For example, if the predicted network has an edge between gene A and gene B, and the GO network also has an edge between gene A and B with a weight of X, then the intersect score would increase by X. If two genes share more than one GO term, then the largest weight is used for the edge. To avoid edges between very loosely associated genes, only GO terms with less than 1000 genes were used for this analysis. The resulting network provides the relationship between genes that share some level of known functionality.

To generate the null distribution of intersect scores for each thresholded PEN, the node labels of each network were randomly permuted 1000 times, and each random network was scored against the GO network. From these null distributions, *t*-statistics were calculated for the score of each thresholded Predictive Expression Network.

For comparison purposes, the noted GO scoring process was also done on a Pearson’s Correlation-generated network, creating co-expression scores. The input data was the same expression data, and Pearson’s Correlation was calculated for each pair of genes. The top 0.1% of correlation values was then analyzed with the GO scoring process, and compared to the top 0.1% of edge scores from the PEN.

### 2.6. Comparison of R to C++ Code

To compare the original iRF R code to the new implementation, both were run on a single node of Summit with a variety of running parameters. These parameters included all combinations of 100, 1000, and 5000 trees and 1, 2, 3, and 4 threads for 1000 features. All combinations were run three times and the scores were averaged. Due to the R code’s doParallel [20] back-end not being designed for an HPC system, the R code was limited to a single CPU on a node with up to four independent threads. For consistency, the new implementation was limited to the same resources. A subset of the *Populus trichocarpa* expression data mentioned above was used as the feature set.

### 2.7. Computational Resources

The computational resources used in this work were Summit, Titan, and a 2015 MacBook Pro laptop. Summit is an Oak Ridge Leadership Computing Facility (OLCF) supercomputer located at Oak Ridge National Laboratory (ORNL). It is an IBM system with approximately 4600 nodes, each with two IBM POWER9 processors, each with 22 cores (176 hardware threads) and six NVIDIA VOLTAV100 GPUs, and 512 GB of DDR4 memory. Titan was a former OLCF supercomputer, recently decommission, and was a Cray system that had approximately 18,688 nodes, each with a 16-core AMD Opteron 6274 processor and 32 GB of DDR3 ECC memory. The 2015 MacBook Pro has a 3.1 GHz Intel Core i7 Processor and 16 GB of DDR3 memory.

## 3. Results

### 3.1. Comparison of the R to C++ Code

Previously, the only published iRF code existed as an R library. This library uses R’s ’doParallel’ functionality, generally allowing for multi-core thread parallelism on shared memory CPUs. However, this system did not function on Summit, so our analysis was limited to running differences on a single Summit CPU.

To compare the R code to the C++ code, both programs were run on a single CPU on Summit. While this is a small resource set, it was sufficient in showing trends and making comparisons between the two implementations. Figure 2 shows the time to completion as the number of threads increases. As the number of threads that the runs are spread over increases, both implementations decrease in run time. However, for the 5000 tree runs for 1, 2, and 3 threads the R code was unable to complete in the 2 h time limit set by the Summit system. This is a good indication that these runs were not efficient enough, as the C++ implementation runs were able to complete.Even though the R implementation uses C++ and Fortran functions internally, it is likely that the overhead of using R and the associated I/O bottlenecks significantly impacts total run time. Similarly, as seen in Figure 3, as the number of trees increases, the R code takes significantly longer than the C++ code to complete. Together, these figures show that in a one-on-one comparison using appropriate resources, the C++ implementation is more efficient than the R implementation, and can handle more computations per unit of time.

### 3.2. Scaling Results for Big Data: Arabidopsis thaliana SNPs to Gene Expression

To show how well our implementation of iRF handles large feature sets, a set of 1.7 million SNPs from *Arabidopsis thaliana* was used to predict the expression of gene AT1G01010. Figure 4 shows the times to completion for the four thread quantities tested on Summit (160 threads per compute node). Our implementation was easily able to handle the data set for all thread quantities and finished in reasonable amounts of time. It is worth noting that there were diminishing returns as the number of nodes gets close to 10 (1600 threads) since the number of trees per node at this scale would only be 100 (1000 trees total spread over 10 nodes) and did not use the resources to its fullest potential. For a larger feature set, a larger number of trees would be advised, and a larger number of nodes could be used more efficiently.

To try to determine approximately how long this calculation would take on a standard laptop, multiple smaller runs were completed. Figure 5 shows the run times for multiple feature amounts for the R iRF code on one CPU of a 2015 MacBook Pro laptop. The linear fit, while not perfect, is accurate enough for a rough estimate for larger feature sizes. Using the provided equation, the 1.7 million feature set run on Summit would take approximately 33 days to complete on a laptop, given that the system had enough memory to contain the data set and results, which most standard laptops do not have. When compared to the approximately 40 min required on 5 nodes of Summit, it is easy to see what a difference these resources, and programs that can use them, can make.

### 3.3. Predictive Expression Networks

We used iRF-LOOP to produce Predictive Expression Networks for *Populus trichocarpa*. Figure 6a shows the run time results for the C++ code for all varying numbers of threads and trees, for one of the approximately 40,000 iRF runs within an iRF-LOOP. Figure 6b shows the total run time as the number of threads increases on Summit and Titan for the C++ code, showing that iRF works comparably on different system architectures. Due to the architecture differences, Summit nodes can independently run 160 threads simultaneously while Titan nodes could only run 16 threads. Summit (in red) had a harder time with larger data on a single thread, but both systems function well as the number of threads increases to appropriate numbers for general uses cases. The full graph of all parameters comparing time to completion on Summit and Titan is available in Appendix A.

Table 1 shows the number of edges and nodes in the four resulting thresholded PENs, as well as for the co-expression comparison network. The GO network that was generated to analyze the PENs contained 16,836 nodes and 3,274,574 edges. Figure 7 shows a small example of the intersected networks with a calculated intersect score.The iRF-LOOP edges that do not have corresponding GO network edges are important for biological discovery. These edges do not necessarily represent ’wrong’ results, but rather interactions found using iRF that are not listed in the GO network. These edges can be used for hypothesis generation of gene interactions, such as regulation, as well as putative gene functions.

Also shown in Table 1 are the mean and standard deviation for the null distribution of the random permutations for each thresholded PEN and the co-expression comparison network. The null distribution and intersect score (in red) for two of the four PEN networks are shown in Figure 8. The null distribution and intersect score comparison between the top 0.1% PEN and 0.1% co-expression network is shown in Figure 9. The other two PEN null distributions are available in Appendix A. The *t*-statistic for each network was calculated from these values, giving the values shown. All PEN *t*-statistic values had a *p*-value of effectively zero, as the iRF intersect scores were all significantly larger than the null distributions. The *t*-statistic for the co-expression comparison network also had a small *p*-value, but it is worth noting that even with more edges than the comparable PEN network, it had a lower GO score. This result confirms that the PENs created using iRF-LOOP are finding more biologically annotated GO edges than would be found by chance., with respect to known GO annotations. Quantile-quantile plots for each of the PEN null distributions are provided in Appendix A, showing that the null distributions are all close enough to normal.

## 4. Discussion

We have presented a high-performance computing-capable implementation of Iterative Random Forest. This implementation uses Ranger, C++, and MPI to use the resources available on multi-node computation resources. We have shown that our implementation can perform X-AI-based eQTL-type analyses with millions of SNPs and have shown its ability to scale with multiple parameters. Using iRF to complete a whole analysis of the 24,175 gene expression profiles for *Arabidopsis thaliana*, assuming each run took approximately the same time as the shown above, would take approximately 705 days using 5 nodes, or 84,680 compute node hours, or 18 h using 4600 Summit nodes. To complete the analysis for all 24,175 gene expression predictions on a laptop using the R code would take approximately 2191 years. While this comparison seems impressive, it should also be noted that for a larger feature set and larger number of trees the large resources will be even more appropriate as the scaling factor will be even less effected by overhead. For cases where the number of SNPs is 10 million and higher, the code should be even more efficient on HPC systems. However, as not everyone has access to the fastest computers in the world this code could still run efficiently on a smaller system using a smaller set of SNPs.

Using this new implementation, we developed iRF-LOOP and used it to produce Predictive Expression Networks which were shown to have biologically relevant information. The process of iRF-LOOP has the potential to be used for a wide variety of data analysis problems. With an appropriate amount of compute resources, it would be possible to build connected networks for each level of ’omics data available for a given species. The same concept could also be used to connect ’omics layers to each other as shown by using the SNPs in the X-AI-based eQTL analysis. This machine learning method is not limited to genetics or biology and has uses in other fields where systems can be represented as a matrix.

Downstream of any iRF analysis, there is the possibility of finding epistatic interactions among features from the resulting forests, using Random Intersection Trees (RIT) [21]. This method works regardless of the data type for the features, where it can find sets of SNPs that influence gene expression or sets of genes that influence other gene’s expression, adding another set of nodes and groups to a Predictive Expression Network.

## 5. Software Availability

The Ranger-based Iterative Random Forest code is available at https://github.com/Jromero1208/RangerBasediRF.

## Figures and Tables

**Figure 1 genes-10-00996-f001:**
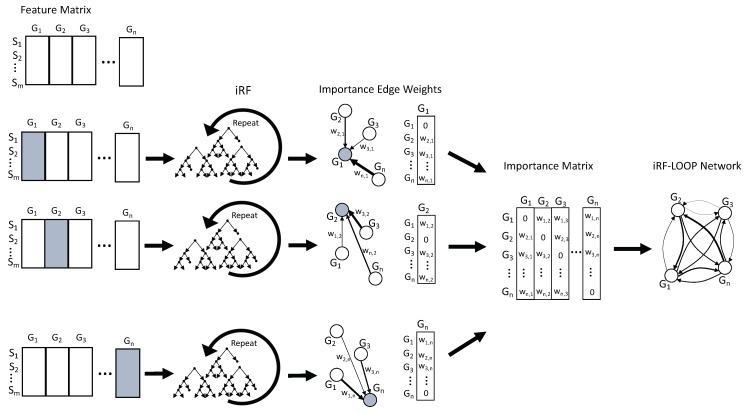
The diagram shows the process of iRF-LOOP for a set of Expression profiles, creating a Predictive Expression Network. Each gene is independently treated as the target for an iRF run, with all other genes as predictors. iRF provides importance scores of each predictor gene, and creates network edge weights between target and predictors. These importance scores are then combined into an edge matrix, providing a value for each possible connection, from which a network can be generated. Generally, the weights are thresholded at some value, determined through other means, and only edges with large enough weights are included in the final network. Due to the inherent directionality of a prediction, the edges are weighted, and not likely to be symmetric.

**Figure 2 genes-10-00996-f002:**
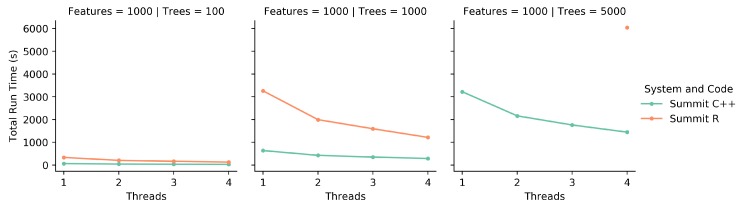
Each of these graphs shows the total run time as the number of threads increases. Both the R code and C++ code were run on Summit. Note for 5000 trees, the R implementation failed to complete using less than 4 threads.

**Figure 3 genes-10-00996-f003:**
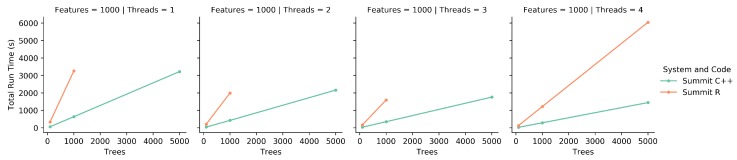
These graphs show a different orientation of the data from Figure 2. Each graph shows the total run time as the number of trees increases, while the number of features and number of threads stays constant. Due to the 5000 tree runs not completing with the R code for 1, 2, or 3 threads, those graphs are missing points.

**Figure 4 genes-10-00996-f004:**
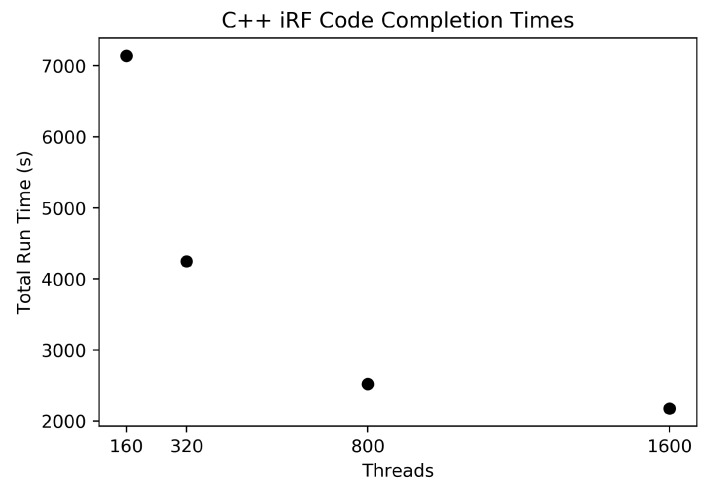
The graph shows the run times for four different compute node quantities, each completing 1000 trees for the 1.7 million SNPs.

**Figure 5 genes-10-00996-f005:**
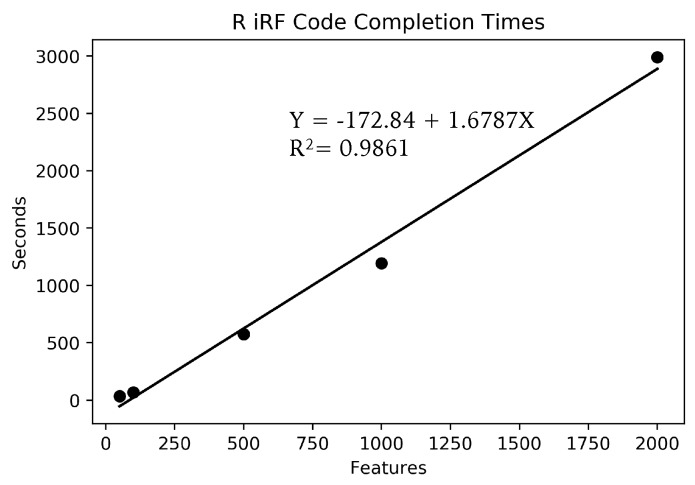
The graph shows the run time for five different feature sizes, on a single CPU of a standard MacBook Pro laptop. Each point represents the average of three runs. A linear regression was fit, with the equation shown. The fit is not perfect, but is enough to indicate that the run time increase approximately linearly in comparison to the number of features.

**Figure 6 genes-10-00996-f006:**
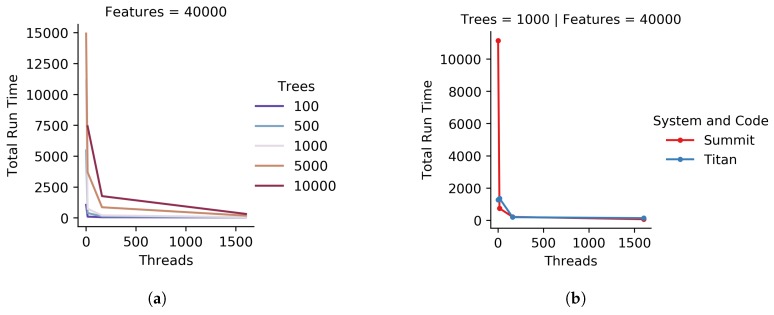
Graph (**a**) provides the total run time for the C++ code on Summit, with various tree and thread counts, for 40,000 features. Graph (**b**) provides a comparison of the C++ code on Summit and Titan, two HPC systems. For both graphs, run time is in seconds.

**Figure 7 genes-10-00996-f007:**
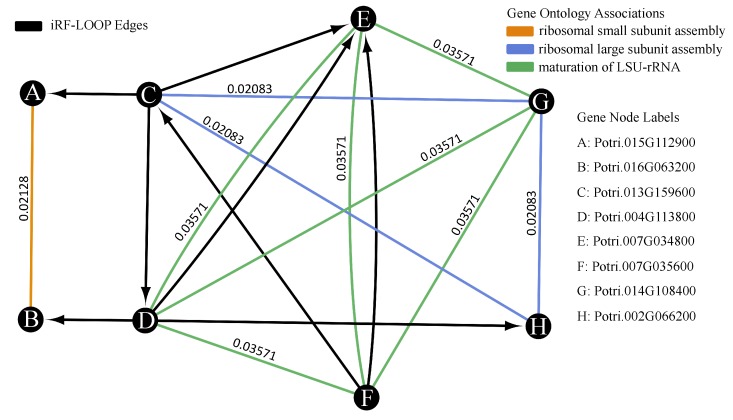
The network shown is a small example from the iRF prediction expression network overlaid with the GO process network. The nodes represent the genes. The black edges represent the iRF edges, which are directed *from* the feature *to* the predicted target. The colored edges represent different GO associations between genes, meaning that they share a GO term. Using the provided GO edge weights, this network has an intersect score of 0.0714, from connections DE and FE with both iRF edges and GO edge.

**Figure 8 genes-10-00996-f008:**
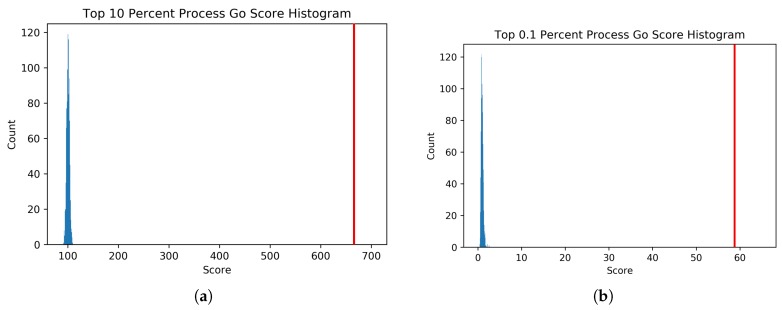
Graph (**a**) shows the null distribution histogram (blue) and the iRF network score (red) for the top 10% of edges. Graph (**b**) shows the null distribution histogram (blue) and the iRF network score (red) for the top 0.1% of edges. Please note that the x-axis is different for the two graphs. Each distribution was calculated from 1000 random permutations.

**Figure 9 genes-10-00996-f009:**
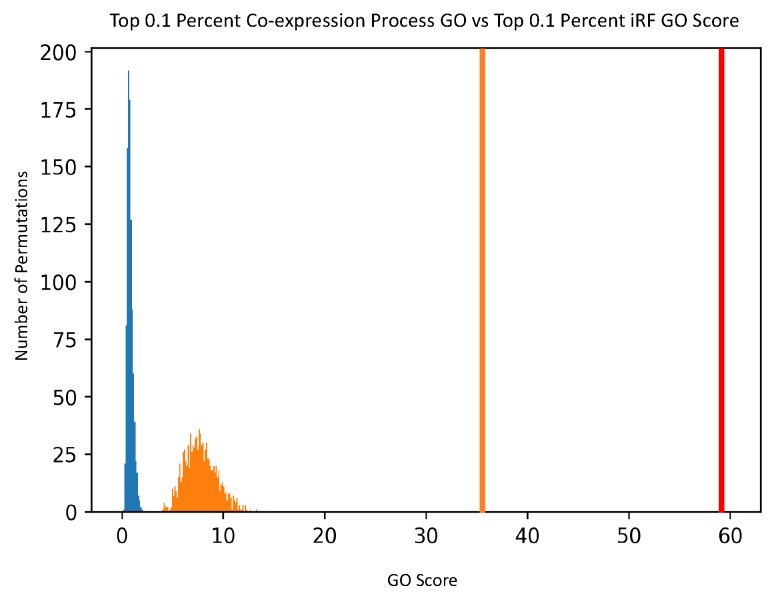
The null distribution histogram of the iRF network is shown in blue, with the network score in red. The co-expression null distribution is shown in orange, with the corresponding network score also in orange.

**Table 1 genes-10-00996-t001:** The table provides the graph results for the 4 thresholded Predictive Expression Networks, as well as the co-expression comparison network. The listed mean and standard deviation are for the corresponding null distributions, as pictured in Figure 8, for the PEN networks. The *p*-values for the listed *t*-statistics were effectively zero.

Network	Nodes	Edges	Intersect Score	Null Dist Mean	Null Dist s.d.	*t*-Statistic
0.1% PEN	26,617	57,112	59.74	0.9831	0.2597	226.27
1% PEN	38,758	563,887	213.28	9.6930	0.8720	233.47
5% PEN	39,349	2,795,636	484.07	48.1309	2.0784	209.74
10% PEN	39,349	5,846,200	692.08	100.5038	2.9316	201.79
0.1% COEX	6261	312,030	34.91	7.7701	1.5668	17.32

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
