# Peer review of "A High-Performance Computing Implementation of Iterative Random Forest for the Creation of Predictive Expression Networks"

_genes, 2019, doi:10.3390/genes10120996_

Round 1

Reviewer 1 Report

pdf attached

Reviewer 2 Report

The authors developed a fast implementation of iterative random forest (iRF) as well as its parallel computing version based on an open-source C++ code. I feel this paper is easy to follow and is practically useful, especially for large-scale industrial applications. I only have two concerns:

1. Indeed, the currently only available open-source implementation for iRF is based on R, and it suffers from lots of limitations, e.g., (1) hard to generalize to large-scale data sets; (2) unable to be deployed in distributed systems. The authors address both limitations. However, is it fair to compare a R implementation with a C++ implementation? At least, the speed improvement may be incurred by many factors.

2. The authors mentioned that the iRF can automatically determine the importance of features or even infer their interactions. On the other hand, the authors also mentioned that the Pearson Correlation coefficient, the mutual information, the sequential feature selection, are able to do the same thing. Is it possible for authors to perform a simple comparison with the correlation coefficient or even the mutual information based ones? Such a comparison can gain more insights on the pros and cons of iRF and the new C++ implementation.

Finally, it should be pointed out that most of the mutual information based feature selection methods belong to the sequential feature selection algorithm family. See examples:
[1] Battiti, Roberto. "Using mutual information for selecting features in supervised neural net learning." IEEE Transactions on neural networks 5, no. 4 (1994): 537-550.
[2] Yu, Shujian, Luis Gonzalo Sanchez Giraldo, Robert Jenssen, and Jose C. Principe. "Multivariate Extension of Matrix-based Renyi's α-order Entropy Functional." IEEE Transactions on Pattern Analysis and Machine Intelligence (2019).

Round 2

Reviewer 2 Report

The authors well addressed all my concerns. I recommend acceptance of this manuscript.